# A Colorimetric Enzyme-Linked Immunosorbent Assay with CuO Nanoparticles as Signal Labels Based on the Growth of Gold Nanoparticles In Situ

**DOI:** 10.3390/nano9010004

**Published:** 2018-12-20

**Authors:** Dehua Deng, Yuanqiang Hao, Jiajia Xue, Xiuhua Liu, Xinyue Xu, Lin Liu

**Affiliations:** 1College of Chemistry and Chemical Engineering, Henan University, Kaifeng 475001, Henan, China; ddh@aynu.edu.cn; 2Henan Key Laboratory of Biomolecular Recognition and Sensing, College of Chemistry and Chemical Engineering, Shangqiu Normal University, Shangqiu 476000, Henan, China; haoyuanqiang@aliyun.com; 3Henan Province of Key Laboratory of New Optoelectronic Functional Materials, Anyang Normal University, Anyang 455000, Henan, China; xjj18239553249@aliyun.com (J.X.); xinyuexu@aliyun.com (X.X.)

**Keywords:** colorimetric immunoassay, CuO nanoparticles, gold nanoparticles, ascorbic acid, fluorescence immunoassay

## Abstract

A colorimetric immunoassay has been reported for prostate-specific antigen (PSA) detection with CuO nanoparticles (CuO NPs) as signal labels. The method is based on Cu^2+^-catalyzed oxidation of ascorbic acid (AA) by O_2_ to depress the formation of colored gold nanoparticles (AuNPs). Specifically, HAuCl_4_ can be reduced by AA to produce AuNPs in situ. In the presence of target, CuO NPs-labeled antibodies were captured via the sandwich-type immunoreaction. After dissolving CuO nanoparticles with acid, the released Cu^2+^ catalyzed the oxidation of AA by O_2_, thus depressing the generation of AuNPs. To demonstrate the accuracy of the colorimetric assay, the released Cu^2+^ was further determined by a fluorescence probe. The colorimetric immunoassay shows a linear relationship for PSA detection in the range of 0.1~10 ng/mL. The detection limit of 0.05 ng/mL is comparable to that obtained by other CuO NPs-based methods. The high throughput, simplicity, and sensitivity of the proposed colorimetric immunoassay exhibited good applicability for assays of serum samples.

## 1. Introduction

Biosensors have been developed for detection of various analytes in the fields of clinical diagnostics, food industry, pharmaceutical chemistry, and environmental science. As to the recognition elements, antibodies are the most commonly used biorecognition molecules in construction of biosensors although many efforts have being made to replace antibodies with alternative recognition molecules [1,2,3]. Thus, immunoassays are still the most widespread analytical methods for the selective and sensitive detection of targets. For example, enzyme-linked immunosorbent assay (ELISA) represents the most popular technique of immunoassays in many fields. However, there still remain some disadvantages about classical ELISA assays, including the complicated and time-consuming implementation procedure, the use of enzyme-labeled, fluorescent or chemiluminescent antibodies, and the bulky measurement instrument. Therefore, numerous attempts have being made to improve the conventional immunosensing concepts [4].

To replace the classic enzyme labels, nanomaterials as signal reporters have attracted tremendous attention in the development of immunosensors, which include metal or metallic oxides, metal-organic frameworks (MOFs), luminescent nanocrystals, etc. [5,6,7,8] After being specifically captured on the sensing interface, the nanolabels can produce a detectable signal directly or be converted into the respective metal ions that can be determined by electric or optical techniques. Since each nanolabel contains large numbers of detectable atoms, the latter is more promising for the construction of highly sensitive immunosensors [9]. For example, CuO nanoparticles have been recently employed for the signal probes of immunosensors because of their advantages of low cost and good stability. After dissolving CuO nanoparticles with acid, the released Cu^2+^ ions can be determined by electric or optical techniques [10,11,12,13,14,15,16]. Among them, the fluorescence assays show high sensitivity. The released Cu^2+^ ions can be directly quantified with fluorescent dyes, quantum dots and nanomaterials or be indirectly determined based on the copper-catalyzed generation of fluorescent molecules [10,11,12,13,14]. In contrast to fluorescence assays, colorimetric assays exhibit high simplicity and require minimum instrumental investment despite their comparatively low sensitivity [17,18,19]. For example, based on the Cu^+^-catalyzed click chemistry, Qu et al. reported a colorimetric immunoassay using azide- and alkyne-modified gold nanoparticles (AuNPs) as the probes [18]. In view of the peroxidase-like catalytic activity of Cu^2+^ to catalyze H_2_O_2_-mediated oxidation of 3, 3’, 5, 5’-tetramethylbenzidine (TMB), Zheng et al. developed an immunosensor by monitoring the generation of colored oxidation product of TMB [19]. The signal has been amplified by the Cu^2+^/Cu^+^-catalyzed reaction. However, in the AuNPs-based immunoassay, the AuNPs need to be prepared and modified with double recognition elements. For the Cu^2+^-catalyzed oxidation of TMB system, high concentration of Cu^2+^ is required to produce colored products. Therefore, there still remains room to develop simple and sensitive colorimetric immunosensors with CuO NPs labels.

Free Cu^2+^ ions can catalyze the oxidation of ascorbic acid (AA) by O_2_; AA as a reducing regent can reduce HAuCl_4_ into AuNPs [20,21]. Based on these facts, we have developed a protease biosensor in that peptide with an amino terminal copper and nickel-binding (ATCUN) motif can inhibit the Cu^2+^-catalytic reaction by complexation with Cu^2+^ to allow for the AA-regulated growth of AuNPs in situ [22]. In view of the high extinction coefficient of AuNPs, herein, we developed an immunosensor by monitoring the generation of AuNPs, which is mediated by the Cu^2+^-catalytic oxidation of AA. Moreover, ATCUN peptide binds to Cu^2+^ with high affinity (10^−16^ M), and Cu^2+^ can cause the fluorescent quenching of fluorophore by electron or energy transfer when binding to the recognition unit [23,24,25]. The released Cu^2+^ ions from the CuO NPs labels were further quantified by a fluorescently-labeled ATCUN peptide probe. The analytical performances of the colorimetric and fluorescent methods were compared with those achieved by other CuO NPs-based immunosensors.

## 2. Materials and Methods

### 2.1. Chemicals and Materials

Prostate-specific antigen (PSA) antigen, first and second PSA antibodies (Ab1 and Ab2) and ELISA kits were ordered from Linc-Bio Science Co. LTD (Shanghai, China). CuO NPs were obtained from Nanjing XFNANO Materials Tech Co., Ltd. (Nanjing, China). Bovine serum albumin (BSA), immunoglobin G (IgG), albumin and hemoglobin were obtained from Sigma-Aldrich (Shanghai, China). Fluorescently-labeled peptide SGHK-Dns was ordered from ChinaPeptides Co., Ltd. (Shanghai, China). Serum sample from one 36-year-old donor was provided by the health center of Anyang Normal University (Anyang, China). All the other chemicals were purchased from Aladdin Reagent Company (Shanghai, China). They are of analytical grade and used as received without additional purification. The solutions were prepared freshly with deionized water treated using a Millipore Milli-Qwater.

### 2.2. Labeling of Antibody with CuO NPs

The procedure for labeling of antibody with CuO NPs follows that of the previously reported methods with slight modification [14,18]. Briefly, 0.5 mg of CuO NPs were dispersed in 0.5 mL of 10 mM phosphate buffer saline (PBS, pH 7.4) by 10-min sonication. Then, 10 μL of the Ab2 (1 mg/mL) was added to the CuO NPs solution. The suspension was shaken slightly for 120 min, followed by centrifugation at 5000 rpm for 5 min. The precipitate was then washed three times with PBS to remove unlabeled Ab2. The resulting CuO NPs-antibody conjugates (Ab2-CuO NPs) were dispersed with 0.2 mL of PBS containing 0.1% BSA and shaken slightly for 30 min. The mixture was then centrifuged and washed to remove free BSA. Finally, the obtained Ab_2_-CuO NPs were re-dispersed in 1 mL of PBS and stored at 4 °C for use.

### 2.3. Procedure for PSA Detection

40 μL of PBS or serum sample containing PSA was added to the ELISA plate and incubated at 37 °C for 1 h. After washing the plate with the diluted cleaning solution five times, 40 μL of the prepared Ab2-CuO NPs suspension was added and incubated for 1 h again. This step was followed by washing the plate with deionized water five times to remove unbound Ab2-CuO NPs. Next, 100 μL of 10 mM HCl was added to the plate to shake for 5 min. To prevent the generated AuNPs from adhering on the plate, the colorimetric assay was conducted on a centrifuge tube. Briefly, 100 μL of PBS (20 mM, pH 7.2) was first added to the above plate to bring the final pH to about 7.0. Then, 50 μL of 0.75 mM AA stock solution was added to the plate for 30-min incubation. Next, 200 μL of the solution was taken out and mixed with 25 μL of 5 mM hexadecyltrimethylammonium chloride (CTAC) solution, and then 25 μL of 2 mM HAuCl_4_ was added to the mixed solution in batches. The color change was observed by the naked eye and the absorption spectra were collected on a Cary 60 UV-Vis spectrophotometer. 

For the fluorescent assay of the released Cu^2+^, 100 μL of PBS containing 4 μM probe was added to the plate following the addition of HCl. Then, the solution was taken out and measured on the FLS980 Steady State Fluorescence and Phosphorescence Lifetime Spectrometer with an excitation wavelength of 340 nm.

### 2.4. Assay of PSA with ELISA Kits

40 μL of standard PSA sample or serum sample was added to the ELISA plate and incubated at 37 °C for 1 h. After washing the plate with the diluted cleaning solution five times, 100 μL of the HRP-labeled antibody (HRP-Ab) solution was added and incubated for 1 h again. Then, the plate was washed with the diluted cleaning solution five times to remove unbound HRP-Ab. Next, 100 μL of the mixed solution of TMB and H_2_O_2_ was added to the plate to incubated for 15 min, which is followed by the addition of 50 μL of stopping solution. Finally, the signal intensity at 450 nm was recorded with an ELISA microplate reader.

## 3. Results and Discussion

### 3.1. Detection Principle

The detection principle of our immunosensor with antibodies-modified CuO NPs as labels is presented in Figure 1, which follows the classical sandwich structure. PSA, a biomarker of prostate diseases such as prostate cancer, prostatitis and benign prostatic hyperplasia, is tested as the model analyte. The immunoassay was carried out on a solid/liquid interface of 96-well plate. The secondary anti-PSA was labeled with CuO NPs (Ab2-CuO NPs). In the detection step, the captured CuO NPs are first dissolved by HCl to produce a lot of Cu^2+^ ions (Figure 1a). AA can be directly oxidized into dehydroascorbic acid (DA) by Cu^2+^; then, the resulting Cu^+^ is rapidly oxidized into Cu^2+^ by O_2_. This Cu^2+^-initiated redox cycling promotes the consumption of AA and thus depresses the AA-regulated generation of AuNPs. Because the color change of the generated AuNPs can be readily observed with naked eyes, no instrument is required for the readout. The amount of the released Cu^2+^ or the captured CuO NPs is dependent upon the target concentration. Thus, the solution color and absorbance signal change induced by the Cu^2+^-catalyzed reaction can be used for the quantitative immunoassays. To demonstrate the accuracy and sensitivity of the colorimetric assay, the released Cu^2+^ was simultaneously quantified with a fluorescence probe. Because Cu^2+^ exhibits both high binding affinity to ATCUN peptide and high quenching ability towards the Dns group, an ATCUN peptide of SGHK labeled with a fluorophore Dns on the side chain of lysine (K) residue was used as the Cu^2+^ sensing probe (denoted as SGHK-Dns) (Figure 1b). After formation of the Cu^2+^-peptide complex, the fluorescence of SGHK-Dns would be quenched.

### 3.2. Optimization of Experimental Conditions

AA concentration and solution pH play decisive roles in the generation of AuNPs. Our early investigations have demonstrated that HAuCl_4_ can be reduced to AuNPs by AA at neutral pH and Cu^2+^ at micromolar concentration can catalyze the exhaustion of 200 μM AA within 10 min [22]. In the present work, the optimized experimental conditions for AuNPs generation followed those of our early work. To demonstrate the binding stoichiometry and fluorescence quenching efficiency of Cu^2+^ to the peptide probe, the fluorescence spectra of SGHK-Dns in the presence of various concentrations of Cu^2+^ were collected. As shown in Figure 2a, the fluorescence signal of the peptide decreased gradually with the increase of Cu^2+^ concentration. The value reached to the minimum in the presence of 1 equiv of Cu^2+^, which is indicative of a 1:1 binding ratio (Figure 2b). The result also demonstrated that labeling of Dns group on the side chain of lysine residue did not decrease the binding affinity of ATCUN peptide with Cu^2+^. Moreover, the quenching efficiency was calculated to be 84.8% with the formula (1 − F’/F^0^) × 100%, where F’ and F^0^ represent the fluorescence intensity of the probe at 552 nm with and without addition of Cu^2+^, respectively. The high quenching efficiency can be attributed the strong interaction and high quenching ability of Cu^2+^ to the fluorescently-labeled peptide. Thus, the designed peptide probe can be used for the sensitive detection of Cu^2+^.

### 3.3. Feasibility

The feasibility of our strategy was investigated by monitoring the change in color and absorption spectra of the detection solutions at various conditions. As shown in Figure 3a, without Ab2-CuO NPs (curve/tube 1) or PSA (curve/tube 2) incubation step, the solution kept red color and the absorbance intensity at 530 nm (A_530_) is high. The absorption peak can be ascribed to the surface plasmon resonance of the generated AuNPs. This result indicated that AA was not oxidized and more AuNPs were generated in the absence of Ab2-CuO NPs or PSA. However, in the PSA detection system, the solution was colourless and the absorbance intensity was greatly decreased, indicating that no or less AuNPs were generated (curve/tube 3). This result indicated that the change of solution color and absorbance intensity is dependent upon PSA capture and the specific antibody-antigen interaction. To further prove that the signal change of the colorimetric assay is caused by the released Cu^2+^ ions from Ab2-CuO NPs, fluorescence assays with SGHK-Dns as the Cu^2+^ detection probe were carried out. As shown in Figure 3b, the fluorescence signal in the PSA detection system is significantly lower than that in the case without PSA or Ab2-CuO NPs capture. The result is consistent with that achieved by the colorimetric assay. Thus, the proposed method based on the target-induced Cu^2+^ concentration change can be used for development of colorimetric or fluorescence immunosensors.

### 3.4. Sensitivity

The sensitivity of the immunosensor was investigated by monitoring the color and signal change of the detection system in the presence of different concentrations of PSA. As shown in Figure 4, increasing PSA concentration resulted in the color change from red to colourless and the gradual decrease in the absorbance intensity at 530 nm. The A_530_ value decreased linearly with increasing PSA concentration from 0.1 to 10 ng/mL. The linear equation is A_530_ = 0.667 − 0.053[PSA] (ng/mL). The detection limit of this method was estimated to be 0.05 ng/mL by determining the smallest concentration of PSA at which the signal is clearly distinguishable from the background. The reproducibility of this method was evaluated by analyzing three freshly prepared PSA samples at the same concentration. The relative standard deviations (RSDs, shown as the error bars in Figure 4b) for assays of the parallel prepared samples are all less than 10.2%, suggesting acceptable reproducibility of the proposed immunosensor. To demonstrate the accuracy of the colorimetric assay, PSA concentration was also determined by measurement of the released Cu^2+^ with the fluorescence probe. As shown in Figure 5, the fluorescence intensity decreases linearly with the increase of PSA concentration in the range of 0.1 to 10 ng/mL with a calibration equation of F = 422.5 − 22.3[PSA] (ng/mL). The detection limit of the fluorescent method was about 0.1 ng/mL. Interestingly, the colorimetric method exhibits a comparable sensitivity with that of the fluorescent assay. Moreover, the detection limit of the proposed immunosensor is comparable to or even lower than that achieved by other CuO-based immunosensors (Table 1). The high sensitivity of the colorimetric immunoassay can be attributed to the high catalytic activity of Cu^2+^ to AA oxidation, the high extinction coefficient of AuNPs, and the signal amplification of CuO NPs. For the fluorescent immunoassay, the high sensitivity results from the high binding affinity and quenching ability of Cu^2+^ to the peptide probe. We believe that the sensitivity may be improved by using more sensitive ELISA plate.

### 3.5. Selectivity

Since the colorimetric assay shows high sensitivity and does not require advanced instrument for signal readout, the selectivity for PSA detection was investigated by the colorimetric method. As shown in Figure 6, for the tested proteins including PSA, BSA, IgG, albumin and hemoglobin, only PSA system resulted in the color change and significant decrease in the absorbance intensity. No significant difference in the color change and absorbance intensity was observed for the non-specific proteins even at the 10-fold higher concentration. The high selectivity can be attributed to the high specificity of antibody with target protein and the low nonspecific adsorption of Ab2-CuO NPs on the ELISA microwells.

### 3.6. Evaluation of Serum Samples

The detection limit of our colorimetric method is lower than the normal value (<4 ng/mL) of PSA released by a healthy prostate. Therefore, the method is promising to determine PSA in a biological sample. To demonstrate the amenability of our method for clinical application, a serum sample was tested (Figure 7). As a result, the solution color of the detection system remained red and the absorbance intensity was slight lower than that of the background value. The initiated PSA concentration in the serum was found to be 1.86 ± 0.12 ng/mL according to the above established standard curve. The value is lower than 4 ng/mL, demonstrating that the donor is healthy. To prove the accuracy of this assay, standard PSA samples at three given concentrations (2, 4 and 8 ng/mL) were added to the serum sample and then analyzed with the colorimetric method. It can be observed that addition of PSA to the sample made the red of detection solution become lighter, which is accompanied by the decrease in the absorbance intensity. The final PSA contents found in the sample are close to the total concentrations of the initiated and spiked PSA. Actually, we found that the sensor response after spiking of 0.1 ng/mL PSA to the serum is clearly distinguishable from that without addition of PSA, which is indicative of high sensitivity of the method for serum sample assay. To ascertain the correctness of the result, the PSA concentrations were further determined with a commercial ELISA kit. No detectable PSA was found in the serum sample with the ELISA kit, demonstrating that the kit is less sensitive for PSA detection. In the PSA-spiked serum samples, the found PSA contents are close to those achieved by the colorimetric method (Table 2). This result indicated that the presented colorimetric immunosensor is applicable for PSA detection in biological samples and can offer a useful means for clinical investigations.

## 4. Conclusions

A colorimetric immunassay method with CuO NPs as the signal labels was reported for the sensitive detection of PSA. The high sensitivity results from the large number of Cu^2+^ released from the CuO NPs, the high catalytic activity of Cu^2+^ to AA oxidation, and the high extinction coefficient of AuNPs. Moreover, the non-enzymatic colorimetric immunosensor has low cost and satisfactory results for assays of PSA in serum samples. In contrast to the AuNPs aggregation-based colorimetric immunassays with CuO NPs labels [18], our method does not require the preparation of double-modified AuNPs, thus reducing the operational complexity and detection cost. Considering the outstanding advantages and excellent performances of the method, we believe that the CuO NPs-based immunoassays may have wide-ranging applications in the clinical diagnosis.

## Figures and Tables

**Figure 1 nanomaterials-09-00004-f001:**
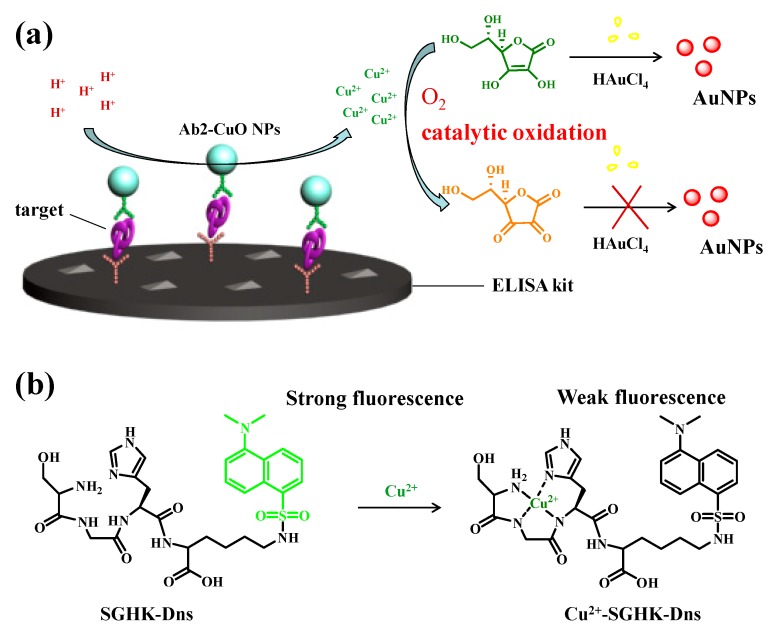
(**a**) Schematic representation of the immunosensor with CuO nanoparticle (CuO NP) labels based on the Cu^2+^-catalyzed ascorbic acid (AA) oxidation and in situ growth of AuNPs; (**b**) Fluorescence detection of the released Cu^2+^ with the probe of SGHK-Dns.

**Figure 2 nanomaterials-09-00004-f002:**
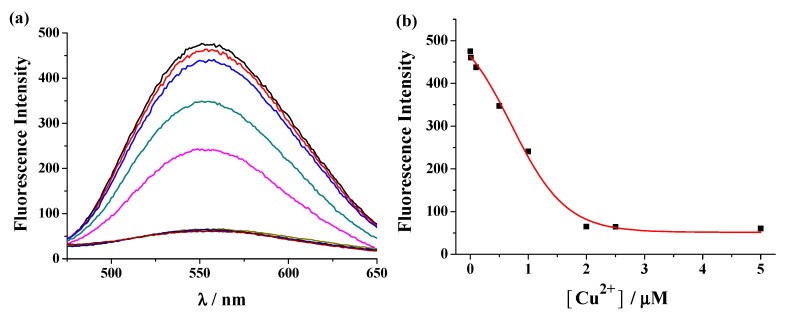
The emission spectra (**a**) and fluorescence intensity (**b**) at 552 nm of 2 μM SGHK-Dns in the presence of different concentrations of Cu^2+^ (from top to bottom: 0, 0.01, 0.1, 0.5, 1, 2, 2.5 and 5 μM).

**Figure 3 nanomaterials-09-00004-f003:**
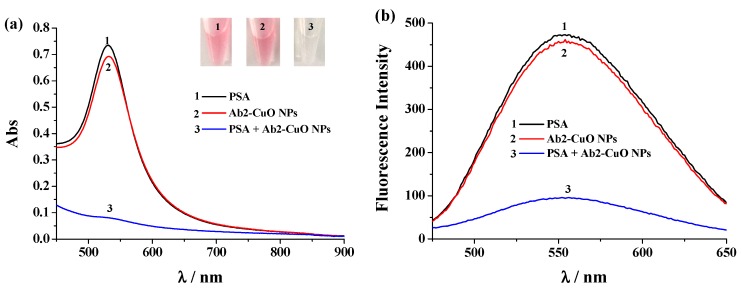
The UV-Vis absorption spectra and photographic images (the inset) (**a**) and emission spectra (**b**) for different detection systems. Curve or tube 1, prostate-specific antigen (PSA); curve 2 or tube 2, Ab2-CuO NPs, PSA; curve 3 or tube 3, PSA + Ab2-CuO NPs. The final concentration of PSA was 50 ng/mL.

**Figure 4 nanomaterials-09-00004-f004:**
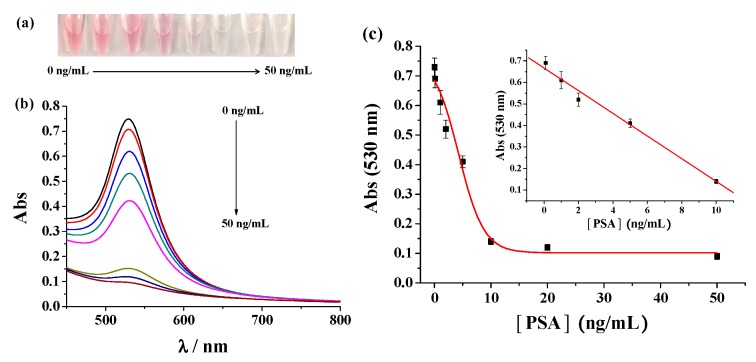
The photographic images (**a**) and UV-Vis absorption spectra (**b**) for assays of different concentrations of PSA (0, 0.1, 1, 2, 5, 10, 20 and 50 ng/mL).

**Figure 5 nanomaterials-09-00004-f005:**
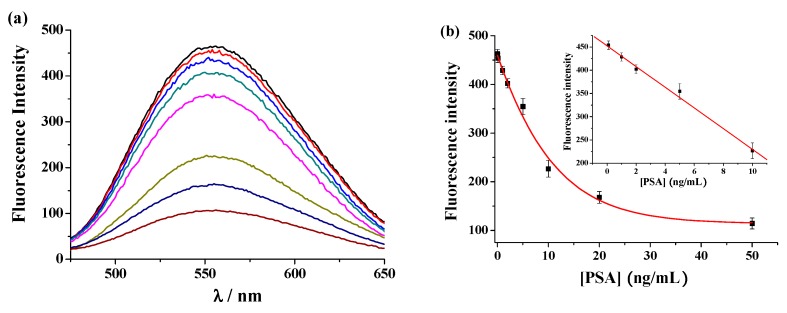
The emission spectra (**a**) and fluorescence intensity (**b**) for assays of different concentrations of PSA (from top to bottom: 0, 0.1, 1, 2, 5, 10, 20 and 50 ng/mL).

**Figure 6 nanomaterials-09-00004-f006:**
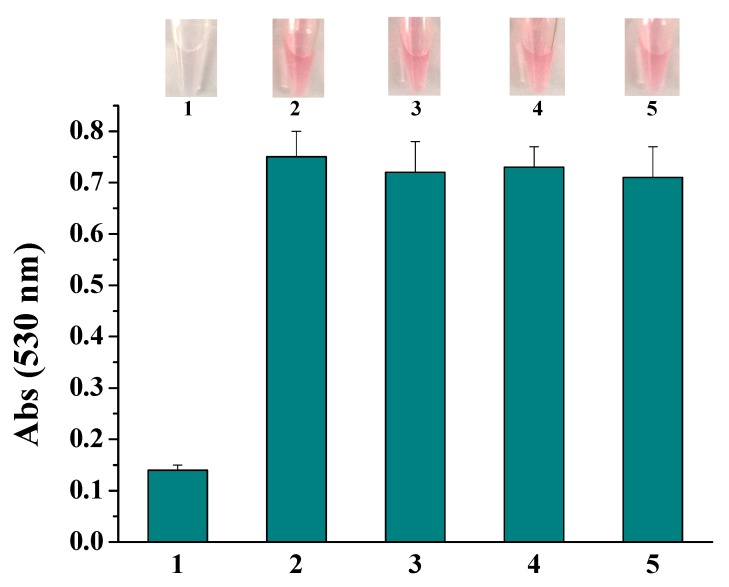
The photographic images and absorption intensity for assays of various proteins (1, PSA; 2, BSA; 3, IgG; 4, albumin; 5, hemoglobin). The final concentration of PSA was 10 ng/mL and that of other proteins is 100 ng/mL.

**Figure 7 nanomaterials-09-00004-f007:**
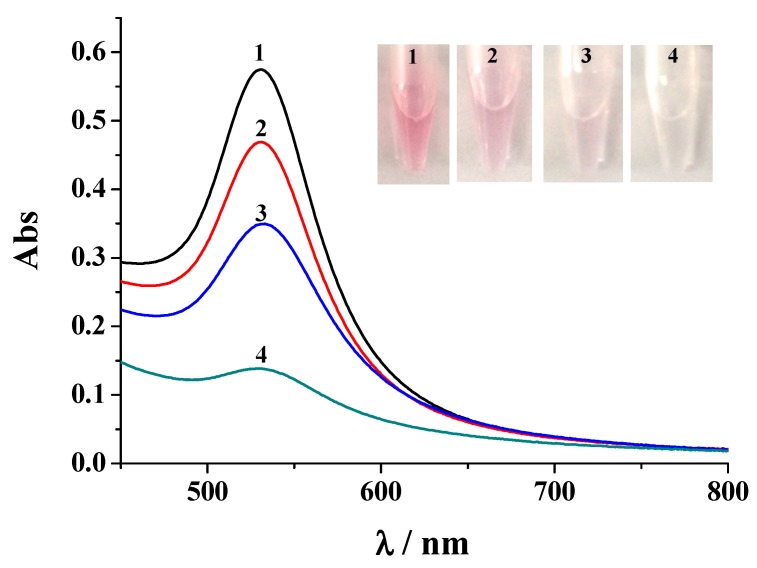
The UV-Vis absorption spectra and the photographic images (the inset) for assays of PSA in serum samples. Curves/tubes 1~4 correspond to the serum samples spiked with 0, 2, 4, and 8 ng/mL PSA, respectively.

**Table 1 nanomaterials-09-00004-t001:** Comparison of analytical performances of various CuO NPs-based biosensors.

Method	Target	Probe for Cu^2+^ Detection	Detection Limit	Linear Range	Ref.
Colorimetry	HIV	AuNPs	Not reported	Not reported	[18]
Colorimetry	CEA	DPHE	26 pg/mL	0.05~100 ng/mL.	[17]
Colorimetry	Glypican-3	TMB/H_2_O_2_	0.26 pg/mL	0.2~200 pg/mL	[19]
Fluorescence	HER2	Quinoxaline derivative	9.65 pg/mL	5~25 pg/mL	[10]
Fluorescence	AFP	triazole complex	0.012 ng/mL	0.025~5.0 ng/mL	[11]
Fluorescence	AFP	CdTe QDs	0.3 pg/mL	0.001~100 ng/mL	[13]
	CA125		0.061 mU/mL	0.0002~100 U/mL	
	CA 153		0.29 mU/mL	0.0001~200 U/mL	
	CEA		1.4 pg/mL	0.005~200 ng/mL	
Fluorescence	AFP	ENFFs	8.3 pg/mL	0.01~200 ng/mL	[12]
Fluorescence	AFP	CdS QDs	0.45 ng/mL,	1~80 ng/mL	[14]
Fluorescence	Exosome	CuNPs	4.8 × 10^4^ particles/μL	7.5 × 10^4^~1.5 × 10^7^ particles/μL	[16]
DPV	H1N1 influenza virus	GCE	10^−12^ g/mL	10^−11^~10^−5^ g/mL	[26]
AAS	IgG		0.19 ng/mL	1~10^4^ g/mL	[15]
Colorimetry	PSA	AuNPs	0.05 ng/mL	0.1~10 ng/mL	This work
Fluorescence	PSA	SGHK-Dns	0.1 ng/mL	0.1~10 ng/mL	This work

HIV, human immunodeficiency virus; CEA, carcinoembryonic antigen; DPHE, 1,2-diphenyl-2-(2-(pyridin-2-yl)hydrazono)ethanone; HER2, human epidermal growth factor receptor 2; AFP, alpha-fetoprotein; ENFFs, electrospun nanofibrous films; QDs, quantum dots; DPV, differential pulse voltammetry; GCE, glass carbon electrode; AAS, atomic absorption spectrometry.

**Table 2 nanomaterials-09-00004-t002:** Assays of PSA in serum samples.

Sample No.	Added (ng/mL)	Found (ng/mL)	ELISA (ng/mL)
1	0	1.86 ± 0.12	undetectable
2	2	3.74 ± 0.23	4.14 ± 0.21
3	4	5.66 ± 0.42	5.72 ± 0.52
4	8	9.78 ± 0.74	10.46 ± 0.83

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
