# Peer review of "A Colorimetric Enzyme-Linked Immunosorbent Assay with CuO Nanoparticles as Signal Labels Based on the Growth of Gold Nanoparticles In Situ"

_nanomaterials, 2018, doi:10.3390/nano9010004_

Round 1
Reviewer 1 Report
The paper described the development of a colorimetric immunoassay for prostate-specific antigen (PSA) detection with CuO nanoparticles (CuO NPs) as signal labels.
This work is interesting and can be appropriate for the publication in Nanomaterials. Nevetheless there are some issues that should be addressed before its publication:
1. The description of the conditions of the modifcation of Ab 2 by Cu should be better clarify
2. Authors should describe better how the LOD has been calculated.
3. Authors should also provide the LOQ value in order to better discuss the data obtained for the analysis of spiked samples.
4. The discussion about the sensitvity of the the two methods (uv-vis and fluorescence) should be better clarified.
5.Authors should discuss the obtained results considering that more sensitive ELISA kit can be available on the market.
6. Reproducibility issue should be discussed and clearly stated.
Author Response
We thank the reviewer for his/her positive and constructive comments: “The paper described the development of a colorimetric immunoassay for prostate-specific antigen (PSA) detection with CuO nanoparticles (CuO NPs) as signal labels. This work is interesting and can be appropriate for the publication in Nanomaterials. Nevetheless there are some issues that should be addressed before its publication:”
Comment 1: “The description of the conditions of the modifcation of Ab2 by Cu should be better clarify.”
Response: The antibodies were attached on the surface of CuO NPs by the physical adsorption interaction, which follows the previously reported procedure (Refs. 14 and 28). More details for the experimental conditions have been included in the revised manuscript,
Comment 2: “Authors should describe better how the LOD has been calculated.”
Response: It is a good question. The quantitative assay was measured based on the signal change with increasing concentration of target. As the previous reports, there are two methods for determining the detection limits (Electroanalysis, 2007, 19, 1239–1257). In the first method, the detection limit can be determined by measuring the sensor response to a dilution series and determining the target smallest concentration at which the sensor response is clearly distinguishable from the response to a blank solution. In the second method, the detection limit can be calculated based on the slope of the dose-response curve and the standard deviation of the blank response. In the present work, the detect limit was estimated with the first method. We have added the sentence as follows: “The detection limit of this method was estimated to be 0.05 ng/mL by determining the smallest concentration of PSA at which the signal is clearly distinguishable from the background.”
Comment 3: “Authors should also provide the LOQ value in order to better discuss the data obtained for the analysis of spiked samples.”
Response: Normally, a healthy prostate releases<4 ng/mL PSA into the circulatory system.The initiated PSA concentration in the serum was found to be 1.86 ± 0.12 ng/mL according to the established method. However, the PSA concentrations are often elevated in the presence of prostate cancer or other prostate disorders. Thus, standard PSA samples at three concentrations (2, 4 and 8 ng/mL) were added to the serum sample and then analyzed with the colorimetric method. The final PSA contents found in the sample are close to the total values of the initiated and spiked PSA. Actually, we found that the sensor response after spiking of 0.1 ng/mL PSA to the serum is clearly distinguishable from that without addition of PSA, which is indicative of high sensitivity of the method for serum sample assay. We have added the comment in Part 3.6.
Comment 4: “The discussion about the sensitvity of the two methods (uv-vis and fluorescence) should be better clarified.”
Response: The sensitivity has been clarified in Part 3.4.
Comment 5: “Authors should discuss the obtained results considering that more sensitive ELISA kit can be available on the market.”
Response: We thank the reviewer for his/her suggestion. For future work, we will use more sensitive commercial ELISA kits for investigation. We have added the sentence on Page 11: “We believe that the sensitivity may be improved by using more sensitive ELISA plate.”
Comment 6: “Reproducibility issue should be discussed and clearly stated.”
Response: It is an excellent comment. We have added the sentences in Part 3.4 to discuss the reproducibility: “The reproducibility of this method was evaluated by analyzing three freshly prepared PSA samples at the same concentration. The relative standard deviations (RSDs, shown as the error bars in Fig. 4B) for assays of the parallel prepared samples are all less than 10.2%, suggesting acceptable reproducibility of the proposed immunosensor.”
Reviewer 2 Report
The manuscript reports a sandwich immunosensor for the detection of PSA by using CuO nanoparticles conjugated to the second antibody as labels. After an acid treatment, the released Cu2+ ions promote the ascorbic acid oxidation by atmospheric oxygen, thus preventing the growth of gold nanoparticles from HAuCl4, and a concomitant decrease in the absorbance at 552 nm is recorded. This method is tested in serum samples with reasonably good results. This work is well-conducted and interesting for the audience of Nanomaterials; however, there are minor issues that should be addressed before its publication :
1- How was LOD estimated?
2- According to the authors the sensitivity for the colorimetric assay and for the fluorescent assay are comparable; however, the slopes for the respective calibration curves are significantly different (-0.053 ua mL ng-1 and -22.3 ua mL ng-1. On the basis of which criterion authors state that conclusion?
3- Table 1 should be cited in the text (page 6, line 201).
4- How many assays have been considered to calculate the error bars in figures 4,5, and 6?
5- Regarding abbreviations:
CTAC appears in section 2.3. but it is not defined
At the end of page 3 “Das” is written, do you mean Dsn?
In table 1, differential pulse voltammetry is abbreviated as DPV not as “EC”.
6- The title 3.6. “Real sample assay” is confused/ imprecise. It could be replaced by “evaluation of serum samples” or something equivalent.
7- Please, revise text format in order to keep homogeneity
8- For future work, there are other ELISA kits with lower LOD.
Author Response
We thank the reviewer for his/her positive and constructive comments: “The manuscript reports a sandwich immunosensor for the detection of PSA by using CuO nanoparticles conjugated to the second antibody as labels. After an acid treatment, the released Cu2+ ions promote the ascorbic acid oxidation by atmospheric oxygen, thus preventing the growth of gold nanoparticles from HAuCl4, and a concomitant decrease in the absorbance at 552 nm is recorded. This method is tested in serum samples with reasonably good results. This work is well-conducted and interesting for the audience of Nanomaterials; however, there are minor issues that should be addressed before its publication:”
Comment 1: “How was LOD estimated?”
Response: It is a good question. The quantitative assay was measured based on the signal change with increasing concentration of target. As the previous reports, there are two methods for determining the detection limits (Electroanalysis, 2007, 19, 1239–1257). In the first method, the detection limit can be determined by measuring the sensor response to a dilution series and determining the target smallest concentration at which the sensor response is clearly distinguishable from the response to a blank solution. In the second method, the detection limit can be calculated based on the slope of the dose-response curve and the standard deviation of the blank response. In the present work, the detect limit was obtained with the first method. We have added the sentence as follows: “The detection limit of this method was estimated to be 0.05 ng/mL by determining the smallest concentration of PSA at which the signal is clearly distinguishable from the background.”
Comment 2: “According to the authors the sensitivity for the colorimetric assay and for the fluorescent assay are comparable; however, the slopes for the respective calibration curves are significantly different (-0.053 ua mL ng-1 and -22.3 ua mL ng-1. On the basis of which criterion authors state that conclusion?”
Response: As the response to comment 1, the detection limit was estimated by determining the smallest concentration of PSA at which the signal is clearly distinguishable from the background. The method for evaluating the sensitivity has be discussed in the revised manuscript.
Comment 3: “Table 1 should be cited in the text (page 6, line 201).”
Response: Yes, “Table 1” has been added on Page 6.
Comment 4: “How many assays have been considered to calculate the error bars in figures 4,5, and 6?”
Response: The error bars were obtained by analyzing three freshly prepared PSA samples. We have added the sentences in Part 3.4 to discuss the reproducibility: “The reproducibility of this method was evaluated by analyzing three freshly prepared PSA samples at the same concentration. The relative standard deviations (RSDs, shown as the error bars in Fig. 4B) for assays of the parallel prepared samples are all less than 10.2%, suggesting acceptable reproducibility of the proposed immunosensor.”
Comment 5: “Regarding abbreviations: CTAC appears in section 2.3. but it is not defined At the end of page 3 “Das” is written, do you mean Dsn? In table 1, differential pulse voltammetry is abbreviated as DPV not as “EC”.”
Response: We have checked the abbreviations carefully and revised the mistakes.
Comment 6: “The title 3.6. “Real sample assay” is confused/ imprecise. It could be replaced by “evaluation of serum samples” or something equivalent.”
Response: We have revised “Real sample assay” into “Evaluation of serum samples”.
Comment 7: “Please, revise text format in order to keep homogeneity.”
Response: The format has been revised.
Comment 8: “For future work, there are other ELISA kits with lower LOD.”
Response: We thank the reviewer for his/her suggestion. For future work, we will use more sensitive commercial ELISA kits for investigation.
Reviewer 3 Report
The paper entitled “A Colorimetric Enzyme-Linked Immunosorbent Assay with CuO Nanoparticles as Signal Labels based on the Growth of Gold Nanoparticles in situ presents a practical and general method to detect an analyte in a medium by conjugation of CuO NPs to PSA antigen followed by an Au NPs growth depression. The authors further tested serum samples and were enabled to detect 3.74ng/ml of PSA (although got false positive detection as well, (as depicts in table 2)). A complementary fluorescence method was used, which gave similar detection thresholds.
In general, this concept was already introduced many times before; nevertheless, some new features could lead to future improvements to this research direction. The introduction section of the paper is clear and comprehend and allow the readers to follow the concept easily.
The paper could be accepted for publication in Nanomaterials after minor revisions:
a. The authors should indicate more clearly in the text that false positive response was generated in a sample lacking PSA.
b. The authors should clarify the experimental section. How the CuO NPs attached to the anti-gen? Ratio? After the CuO were dissolved in HCl, what was the used pH? Did the authors changed the pH before moving to the next step? This part (2.2, 2.3) is not clear!
Author Response
We thank the reviewer for his/her positive and constructive comments: “The paper entitled “A Colorimetric Enzyme-Linked Immunosorbent Assay with CuO Nanoparticles as Signal Labels based on the Growth of Gold Nanoparticles in situ” presents a practical and general method to detect an analyte in a medium by conjugation of CuO NPs to PSA antigen followed by an Au NPs growth depression. The authors further tested serum samples and were enabled to detect 3.74 ng/ml of PSA (although got false positive detection as well, (as depicts in table 2)). A complementary fluorescence method was used, which gave similar detection thresholds. In general, this concept was already introduced many times before; nevertheless, some new features could lead to future improvements to this research direction. The introduction section of the paper is clear and comprehend and allow the readers to follow the concept easily. The paper could be accepted for publication in Nanomaterials after minor revisions:”
Comment 1: “The authors should indicate more clearly in the text that false positive response was generated in a sample lacking PSA.”
Response: In this work, PSA was tested as a model protein. The normal value of PSA released by a healthy prostate is below 4 ng/mL. The initiated PSA concentration in the serum without spiking standard PSA sample was found to be 1.86 ± 0.12 ng/mL. The value is lower than 4 ng/mL, demonstrating that the donor is healthy. We have discussed the result in Part 3.6.
Comment 2: “The authors should clarify the experimental section. How the CuO NPs attached to the anti-gen? Ratio? After the CuO were dissolved in HCl, what was the used pH? Did the authors changed the pH before moving to the next step? This part (2.2, 2.3) is not clear!”
Response: The antibodies were attached on the surface of CuO NPs by the physical adsorption interaction, which follows the previously reported procedure (Refs. 14 and 28). More details for the experimental conditions have been included in the revised manuscript, which have been highlighted in
red words (Part 2.2, 2.3).
Reviewer 4 Report
The authors describe colorimetric immunoassay for prostate specific antigen with copper oxide nanoparticles. The particles are dissolved in hydrochloric acid and copper ions catalyse oxidation of ascorbic acid. The assay is stated as a simple and sensitive one.
It seems that similar methods have been already applied for such assay. The authors should state novelty of the method.
Moreover, when the authors want to apply the method in practice then they should mention how cheap it is in comparison with already used assays. The assay must be much better described to receive reproducible results. What about presence of other metal ions in samples?
Author Response
Comment: “The authors describe colorimetric immunoassay for prostate specific antigen with copper oxide nanoparticles. The particles are dissolved in hydrochloric acid and copper ions catalyse oxidation of ascorbic acid. The assay is stated as a simple and sensitive one. It seems that similar methods have been already applied for such assay. The authors should state novelty of the method. Moreover, when the authors want to apply the method in practice then they should mention how cheap it is in comparison with already used assays. The assay must be much better described to receive reproducible results. What about presence of other metal ions in samples?”
Response: We thank the referee for his/her review. CuO nanoparticles have been employed for the signal probes of immunosensors because of their advantages of low cost and good stability. After dissolving CuO nanoparticles with acid, the released Cu2+ ions can be determined by electric or optical techniques. In view of the high extinction coefficient of AuNPs, herein, we developed an immunosensor by monitoring the generation of AuNPs, which is mediated by the Cu2+-catalytic oxidation of AA. The results were further verified by a fluorescent probe. In contrast to the fluorescence assay, colorimetric assays exhibit high simplicity and require minimum instrumental investment. In contrast to the existing AuNPs aggregation-based colorimetric immunassays with CuO NPs labels (Angew. Chem. Int. Ed. 2011, 50, 3442-3445), our method does not require the preparation of double-modified AuNPs, thus reducing the operational complexity and detection cost. The regents for signal readout including HAuCl4, ascorbic acid and CuO are cheap and stable. Additionally, less dosage of reagents were required for each assay because of the high catalytic activity of Cu2+ and the high extinction coefficient of AuNPs. These points have been presented in the main text. The reproducibility of this method was evaluated by analyzing three freshly prepared PSA samples at the same concentration. The relative standard deviations (RSDs, shown as the error bars in Fig. 4B) for assays of the parallel prepared samples are all less than 10.2%, suggesting acceptable reproducibility of the proposed immunosensor (we have added the sentences in Part 3.4). Other redox metal ions may promote the oxidation of AA; in this work, all the solutions were prepared freshly with deionized water treated using a Millipore Milli-Qwater. Using inductively coupledplasma mass spectrometry (ICP-MS), we found that the concentrations of heavy metal ions after such a treatment were below 1 nM. For sample assays, the detection was carried out step-by-step. No extra metal ions were introduced to the detection solution (AA and HAuCl4).
Round 2
Reviewer 4 Report
Thank you for the changes.